# UniAudio: An Audio Foundation Model Toward Universal Audio Generation

## Abstract

Large Language models (LLM) have demonstrated the capability to handle a variety of generative tasks. This paper presents the UniAudio system, which, unlike prior task-specific approaches, leverages LLM techniques to generate multiple types of audio (including speech, sounds, music, and singing) with given input conditions. UniAudio 1) first tokenizes all types of target audio along with other condition modalities, 2) concatenates source-target pair as a single sequence, and 3) performs next-token prediction using LLM. Also, a multi-scale Transformer model is proposed to handle the overly long sequences caused by the residual vector quantization-based neural codec in tokenization. Training of UniAudio is scaled up to 165K hours of audio and 1B parameters, based on all generative tasks, aiming to obtain sufficient prior knowledge not only in the intrinsic properties of audio but also the inter-relationship between audio and other modalities. Therefore, the trained UniAudio model has the potential to become a foundation model for universal audio generation: it shows strong capability in all trained tasks and can seamlessly support new audio generation tasks after simple fine-tuning. Experiments demonstrate that UniAudio achieves state-of-the-art or at least competitive results on most of the 11 audio generation tasks. Demo and code are released.[1] .

## 1 Introduction

Audio generation is an important component of generative AI. Recently, the popularity of generative AI has induced increasingly emergent and varying needs in audio generation: audio is expected to be generated based on humans's demands, such as speech synthesis (TTS), voice conversion (VC), singing voice synthesis (SVS), text-to-sound, and text-to-music. Prior works on audio generation tasks are commonly task-specific: their designs heavily leverage domain knowledge and their usage is restricted to fixed setups (Tan et al., 2021; Luo & Mesgarani, 2019; Zmolikova et al., 2023; Huang et al., 2021b; Cho et al., 2021). Instead of taking care of each task independently, this work is an attempt to achieve universal audio generation, which intends to accomplish multiple audio generation tasks with only one unified model. The universal audio generation model is expected to obtain sufficient prior knowledge in audio and related modalities, which has the potential to provide simple and effective solutions for the increasing needs of generating diverse types of audio.

The superiority of Large Languge Models (LLM) in text-generative tasks inspires a series of LLM-based models in audio generation (Wang et al., 2023a; Kharitonov et al., 2023; Huang et al., 2023b; Agostinelli et al., 2023; Borsos et al., 2023). Among these works, LLM's capability in independent tasks has been extensively studied in tasks like text-to-speech (TTS) (Wang et al., 2023a; Kharitonov et al., 2023; Huang et al., 2023b) and music generation (Agostinelli et al., 2023; Copet et al., 2023), and achieves competitive performance. However, LLM's ability to process multiple tasks with a unified model is less exploited in audio generation research: most existing LLM-based works are still designed for single tasks (Wang et al., 2023a; Kharitonov et al., 2023). We argue that achieving universality and versatility in audio generation through the LLM paradigm is promising but has not yet been comprehensively studied before this work.

Toward universal audio generation, this work presents UniAudio, which adopts LLM techniques and is able to generate multiple types of audio (speech, sounds, music, and singing) conditioned on various input modalities, such as phoneme sequences, textual descriptions, and audio itself. The proposed UniAudio is mainly featured as follows: First, all types of audio, along with all other input

---

[1] `https://uniaudio666.github.io/demo_UniAudio/`
    * Equal contribution; † Corresponding author

modalities, are tokenized as discrete sequences. Specifically, a universal neural codec model is built to effectively tokenize audio regardless of the audio type, and other tokenizers are used to tokenize other different modalites. Then, UniAudio concatenates the source-target pair as a single sequence. Lastly, UniAudio performs next-token prediction using LLM. The residual vector quantization (Zeghidour et al., 2021) based on neural codecs is used in the tokenization process, resulting in overly long token sequences (one frame corresponding to multiple tokens) that cannot be processed efficiently by LLM. A multi-scale Transformer architecture is designed to reduce computational complexity by modeling the inter- and intra-frame correlation separately. Specifically, a global Transformer module is used to model the inter-frame correlation (*e.g.* semantic level), and a local Transformer module is used to model the intra-frame correlation (*e.g.* acoustic level).

To demonstrate the scalability of UniAudio for new tasks, the building process of UniAudio takes two stages. Firstly, the proposed UniAudio is trained on multiple audio generation tasks jointly, which allows the model to obtain sufficient prior knowledge not only of the intrinsic properties of audio but also of the interrelationship between audio and other input modalities. Secondly, through fine-tuning, the trained model can seamlessly support more unseen audio generation tasks. Thus, UniAudio has the potential to become a foundation model for universal audio generation: it is able to continuously support emergent needs in audio generation. Experimentally, our UniAudio supports 11 audio generation tasks: the training stage includes 7 audio generation tasks, while 4 tasks are further added in the fine-tuning stage. The building process of UniAudio is scaled up to 165k hours of audio and 1B parameters. Among the 11 tasks, UniAudio consistently obtains competitive performance in both objective and subjective evaluations. State-of-the-art results are even achieved on most of these tasks. Further investigation suggests that training multiple tasks simultaneously in the training stage is mutually beneficial to each task involved. In addition, UniAudio can effectively adapt to new audio generation tasks and outperform task-specific models with a non-trivial gap.

To sum up, this work reveals that building universal audio generation models is necessary, promising, and beneficial. The main contributions of this work are summarized as follows:
(1) Toward universal audio generation, UniAudio is presented as a unified solution for 11 audio generation tasks.

(2) Per methodology, UniAudio provides novel approaches for (i) sequential representations of audio and other input modalities; (ii) uniform formulation for LLM-based audio generation tasks; and (iii) efficient model architecture specifically designed for audio generation.

(3) Per experiments, the overall performance of UniAudio is well validated, and the benefits of building a versatile audio generation model are verified by exhaustive experimental results.

(4) Demo and code are released, in the hope that UniAudio can become a foundation model that supports emergent audio generation in future research.

## 2 UNIAUDIO

This section introduces the technical details of the proposed UniAudio. Section 2.1 explains how audio and other modalities are tokenized. Then, all considered audio generation tasks are uniformly formulated in Section 2.2. Subsequently, the multi-scale Transformer architecture is proposed in Section 2.3 to handle the overly long sequence challenge caused by the adoption of neural codecs.

### 2.1 TOKENIZATION

LLM are commonly used for sequential modeling, so audio and all other input modalities are tokenized before being processed. These processes for each modality are completed by independent modules. All of these modules are fixed in the optimization of UniAudio or parameter-free.

### 2.1.1 AUDIO

For all audio generation tasks considered in this work, audio, regardless of its types (speech, sounds, music, or singing), is the target to predict. Instead of modeling different types of audio separately, UniAudio intends to tokenize all types of audio as a single and unified modality (even though they commonly have distinct patterns, such as frequency span), which requires a model that is well-suited to mapping all audio types into a shared latent space. Following Wang et al. (2023a); Kharitonov et al. (2023), neural codec models (Défossez et al., 2022; Yang et al., 2023b; Kumar et al., 2023) are used in this work for audio tokenization. An audio signal of duration $d$ with sample rate $f_s$ can be

represented by a sequence $\mathbf{x} \in [-1, 1]^{d * f_s}$. An audio neural codec intends to compress $\mathbf{x}$ and then recover it as $\hat{\mathbf{x}}$ using an encoder-decoder architecture with a quantization module:

$$\mathbf{h} = \text{Encoder}(\mathbf{x}) \in \mathcal{R}^{T*L}; \quad \hat{\mathbf{h}} = \text{Quantization}(\mathbf{h}); \quad \hat{\mathbf{x}} = \text{Decoder}(\hat{\mathbf{h}}) \tag{1}$$

where $T$ denotes the number of audio frames after down-sampling in the encoder, and $L$ denotes the feature dimension of the encoder. The discrete representations of audio are the intermediate product of the quantization process. Given any frame of hidden output $\mathbf{h}_t$, the integer vector $\mathbf{z}_t = [z_t^1, ..., z_t^{n_q}]$ is generated by Residual Vector Quantization (RVQ) (Zeghidour et al., 2021), where $n_q$ denotes the number of vector quantization layers. Iteratively, each element $z_t^k$ is the index among all pre-learned and fixed $k$-th level quantizer vectors $\{\mathbf{q}_k^*\}$ that has the smallest L2 distance to the residual between $\mathbf{h}_t$ and the sum of all previous chosen quantizer vectors $\{\mathbf{q}_j^{z_t^j}, j = 1, ..., k-1\}$. With the discrete representation $\mathbf{z}_t$, $\hat{\mathbf{h}}_t$ is reconstructed as a close estimation of $\mathbf{h}_t$ that can be used to recover $\mathbf{x}_t$ with the decoder.

$$z_t^k = \arg\min_m \text{Distance}(\mathbf{h}_t - \sum_{j=1}^{k-1} \mathbf{q}_j^{z_t^j}, \mathbf{q}_k^m); \quad \hat{\mathbf{h}}_t = \sum_{j=1}^{n_q} \mathbf{q}_j^{z_t^j}; \quad 1 \le k \le n_q \tag{2}$$

The discrete representation of all audio frames $\mathbf{z} \in \mathbf{Z}^{T \times n_q}$ is a matrix and needs to be converted into a sequence before being processed by LM: it is simply flattened as a sequence, in which every $n_q$ element for one frame is consecutive. Without specifically stated, we set $n_q = 3$ in our experiments. As the waveform can be recovered from $\mathbf{z}$ with a neural codec decoder, the rest of this paper mainly discusses how to predict the audio token sequence $\mathbf{z}$ using LLM techniques. As UniAudio intends to generate both speech and non-speech content, we build the codec model on our own and with broader data coverage. Details of our codec configuration is in Appendix E.

### 2.1.2 OTHER MODALITIES

Besides audio, other modalities considered in UniAudio also need to be represented as sequences. In addition, most of these sequences are transformed into discrete ones through tokenization. The serialization and tokenization of these input modalities, along with their key features, are briefly summarized as below.

**Phoneme:** Phonemes are the basic units of speech pronunciation in linguistics. Phoneme sequences have multiple sources: (1) when only text is available, phoneme sequence without duration information can be obtained by text-to-phoneme mapping using a pronunciation dictionary; (2) when only speech is available, phoneme sequence with duration information is obtained by beam search of the DNN-HMM system (Hinton et al., 2012); (3) when both text and speech are available, phoneme sequence with duration information is obtained by forced alignment of the DNN-HMM system [2].

**MIDI:** MIDI (Zhang et al., 2022) is widely used for singing voice synthesis tasks. F0 and duration information are included in the MIDI. We use the duration information to flatten the F0 sequence, so that the frame-level F0 sequence is obtained.

**Text:** Text acts as a effective carrier of human instructions in audio generation tasks (Yang et al., 2023a; Copet et al., 2023). In this work, these textual instructions are represented as continuous embeddings derived from pre-trained text LLM (Raffel et al., 2020), as these embeddings contain rich textual semantics. Processing these continuous embeddings with LLM is further clarified in Section 2.3 [3].

**Semantic Token:** The semantic tokens are derived from the continuous embeddings output by audio self-supervised learning (SSL) models. These continuous representations are highly informative and can be adopted in both speech understanding (Rubenstein et al., 2023) and generative tasks (Borsos et al., 2023). Following Huang et al. (2023b), these continuous representations are tokenized by performing K-means clustering (Hsu et al., 2021) over these continuous representations. Since the continuous representations are frame-level, the semantic tokens also encode duration information [4].

---

[2] CMUDict (http://www.speech.cs.cmu.edu/cgi-bin/cmudict) is adopted as the pronunciation dict; kaldi recipe (https://github.com/kaldi-asr/kaldi/tree/master/egs/librispeech/s5/local/chain/run_tdnn.sh) is adopted to build the deep neural network-hidden Markov model (DNN-HMM) system.

[3] The encoder of T5 (https://github.com/google-research/text-to-text-transfer-transformer) is used to extract the continuous text embeddings.

[4] The 9-th layer hidden output of Hubert (Hsu et al., 2021) is adopted as the semantic token representations (https://github.com/facebookresearch/fairseq/hubert). The number of clusters for K-means is 500.

## 2.2 UNIFIED TASK FORMULATION

Table 1: Sequence formats of all tasks supported by UniAudio. Text color represents modality. black: audio; green: phoneme; blue: MIDI; purple: text; brown: semantic token. ♣ means tasks that generate audio with deterministic length. ◇: means tasks that are only included in the fine-tuning stage. The speaker prompt is a 3-second speech and is used to represent the speaker identification.

| Task | Conditions | Audio Target |
|---|---|---|
| Text-to-Speech (TTS) (Wang et al., 2023a) | phoneme, speaker prompt | speech |
| Voice Conversion (VC) ♣ (Wang et al., 2023e) | semantic token, speaker prompt | speech |
| Speech Enhancement (SE) ♣ (Wang et al., 2023b) | noisy speech | speech |
| Target Speech Extraction (TSE) ♣ (Wang et al., 2018) | mixed speech, speaker prompt | speech |
| Singing Voice Synthesis (SVS) (Liu et al., 2022) | phoneme (with duration), speaker prompt, MIDI | singing |
| Text-to-Sound (Sound) (Yang et al., 2023c) | textual description | sounds |
| Text-to-Music (Music) (Agostinelli et al., 2023) | textual description | music |
| Audio Edit (A-Edit) ♣◇ (Wang et al., 2023d) | textual description, original sounds | sounds |
| Speech dereverberation (SD) ♣◇ (Wu et al., 2016) | reverberant speech | speech |
| Instruct TTS (I-TTS)◇ (Guo et al., 2023) | phoneme, textual instruction | speech |
| Speech Edit (S-Edit) ◇ (Tae et al., 2021) | phoneme (with duration), original speech | speech |

For all tasks considered in UniAudio, the target audio is generated based on given conditions. With the same target modality, i.e., audio, it is the conditions that define different audio generation tasks. However, even with the variance in conditions, all tasks can still be uniformly formulated as sequential modeling tasks that can be processed by LLM: both the target audio and the conditions are first transformed as sub-sequences and spliced as [*conditions*, *target*] sequences to be processed.

UniAudio supports 11 audio generation tasks in total. The sequential formats of each task are defined in Table 1, in which the sub-sequences of all modalities are derived as in Section 2.1. However, due to the unique configurations of each task, some of the condition sub-sequences are subject to task-specific pre-processing operations during the tokenization. For audio, these operations are mainly for data corruption, such as adding noise, reverberation, and speech mixed with other speakers in the raw audio before tokenization. For phoneme and semantic tokens, duration information is reserved by default but can also be removed. For singing voice synthesis and speech edit tasks, the duration information of phoneme is used. For TTS and I-TTS tasks, the duration information is not used. For MIDI, the duration information is used repeat the F0 sequence. For text embeddings, no operations are applied in this work.

To avoid ambiguity, some special discrete tokens (enclosed by <>) are inserted to indicate (1) the start and end of the whole sequence; (2) the start and end of each sub-sequence of a certain modality; and (3) the task identifier. For example, for a text-to-sound task sequence that generates target audio based on textual description, the whole sequence is like: *<start> <sound_task> <text_start> text_sequence <text_end> <audio_start> audio_sequence <audio_end> <end>*.

## 2.3 MULTI-SCALE TRANSFORMER

Previous work on LLM-based audio generation (Copet et al., 2023) advocates to modeling the discrete audio tokens as flattened sequences. If so, these sequences are processed in the length of $T \times n_q$, which is highly challenging considering the quadratic space complexity of Transformer (Vaswani et al., 2017) with respect to the lengths. Inspired by Yu et al. (2023), a multi-scale Transformer architecture is specifically designed for discrete audio sequences, which is a hierarchical model that processes the inter- and intra-frame correlation by global and local Transformer modules separately. An overview of the proposed architecture is in Figure 1. Instead of processing the whole flattened sequence token-by-token like prior works (Kharitonov et al., 2023), the multi-scale transformer considers patches (i.e., every consecutive $n_q$ token) as the global modeling units and then handles the tokens within each patch locally. Note that both the global and local Transformers are causal.

For audio token sequences, each patch accounts for $n_q$ consecutive audio tokens that exactly represent one audio frame. First, as suggested in Equation 2, regardless of the exact choices of each quantization vector $\mathbf{q}_*^{z_t^*}$, it is the summed quantization vector $\hat{\mathbf{h}}_t$ that is used to represent the audio frame. Thus, in the embedding stage, each patch (a.k.a., frame) is represented by the summed vector of the corresponding embeddings before entering the global Transformer. Second, the global Transformer is to predict audio frame-by-frame: to predict the frame $\mathbf{x}_t$, it outputs the continuous representations

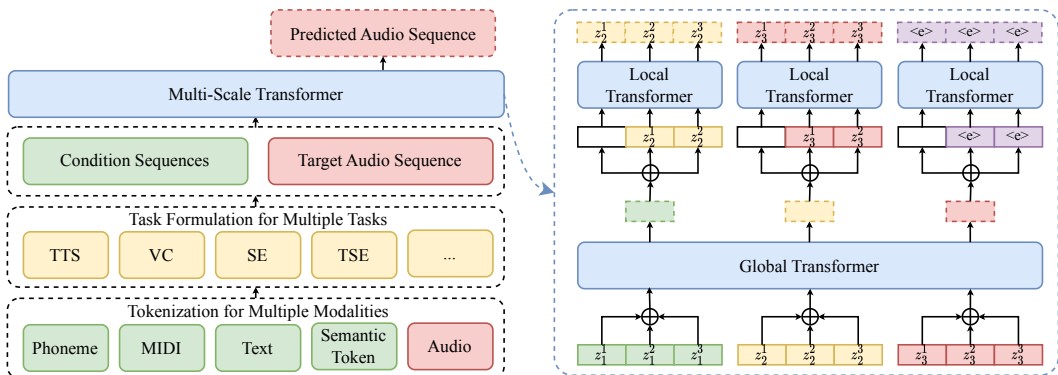

Figure 1: Overview of UniAudio (left) and multi-scale Transformer architecture (right). <e> represent the end of the sequence. $z_t^k$ denotes the k-th audio token at t-th frame.

that include frame $\mathbf{x}_{t-1}$ and all previous content. These continuous representations will be further processed by the local Transformer. Third, also as in Equation 2, given the hidden representation $\mathbf{h}_t$, the acquisition of $\mathbf{z}_t$ is independent of any hidden output other than $\mathbf{h}_t$. Inspired by this, it is reasonable to predict the discrete tokens for frame $\mathbf{x}_t$, a.k.a., patch $\mathbf{z}_t$, only with the hidden output of global Transformer corresponding to frame $\mathbf{x}_{t-1}$. To be more detailed, as the acquisition of each token $z_t^k$ is auto-regressively dependent on its prior tokens $\{z_t^j | j < k\}$, a local Transformer is adopted to predict the patch sequence $\mathbf{z}_t$ in auto-regressive style. During this process, the corresponding vector output by the global transformer acts as a patch-level context, which is linearly transformed and then added to the embedded results of each token $z_t^k$.

The proposed multi-scale Transformer architecture is also compatible with discrete and continuous sequences besides audio. For all discrete tokens except audio (phoneme, semantic, MIDI and special tokens), each token has independent semantics and thus should account for one patch. So these discrete tokens repeat for $n_q$ times to fill each patch. The continuous text embeddings are also repeated for $n_q$ times for the same purpose. Additionally, their embedding process is replaced by a linear transformation while their predicting targets for local Transformer are consecutive special tokens <*continuous_token*>.

The design of the proposed multi-scale Transformer can effectively reduce computational complexity. First, the equivalent sequence length for the global Transformer is reduced from $T \times n_q$ to $T$, which makes the global modeling cost independent to $n_q$ and thus the adoption of a larger $n_q$ becomes feasible. Second, the intra-patch computation to generate the discrete tokens for each frame is offloaded to the local Transformer. The computation on the local transformer is comparatively light since it only processes the very short sequence (fixed to the length of $n_q$) and empirically has fewer parameters than the global Transformer by design.

## 3 EXPERIMENTS

This section first introduces the experimental setup in Section 3.1. The results for the training stage and the fine-tuning stage are presented in Section 3.2 and 3.3 respectively. Ablation studies are presented in Section 3.4.

### 3.1 EXPERIMENTAL SETUP

**Data and Model:** UniAudio is built on labeled datasets. Specifically, 12 datasets are adopted in this work, all of which are publicly available. The overall audio volume is 165K hours. Detailed data statistics and their adoption for each task are in Appendix A.1. Discrete tokens from all modalities form a joint vocabulary of size 4212, including all special tokens. Vanilla Transformer decoder layers with causality are consistently adopted in global and local Transformer. The overall parameter budget is roughly 1B. Detailed model configuration is in Appendix A.2. Existing neural codec models are sub-optimal for universal audio generation, mainly due to data coverage. An improved neural codec model is then built with fewer quantization levels $n_q$, smaller frame-per-second rate, higher quality, and wider coverage (see Appendix E).

**Training and Inference:** The training stage includes 7 tasks while 4 new tasks are added in the fine-tuning stage. Table 1 specifies the tasks for fine-tuning only. Both the training and fine-tuning

Table 2: Performance evaluation for UniAudio and selected prior works in the training stage

| Task | Model | Objective Evaluation | | Subjective Evaluation | |
|---|---|---|---|---|---|
| | | Metrics | Results | Metrics | Results |
| Text-to-Speech | Shen et al. (2023) | SIM($\uparrow$) / WER($\downarrow$) | 0.62 / 2.3 | MOS($\uparrow$) | **3.83**±**0.10** / 3.11±0.10 |
| | UniAudio | | **0.71 / 2.0** | / SMOS($\uparrow$) | 3.81±0.07 / **3.56**±**0.10** |
| Voice Conversion | Wang et al. (2023e) | SIM($\uparrow$) / WER($\downarrow$) | 0.82 / 4.9 | MOS($\uparrow$) | 3.41±0.08 / 3.17±0.09 |
| | UniAudio | | **0.87 / 4.8** | / SMOS($\uparrow$) | **3.54**±**0.07 / 3.56**±**0.07** |
| Speech Enhancement | Richter et al. (2023) | PESQ($\uparrow$) / VISQOL($\uparrow$) / DNSMOS($\uparrow$) | **3.21 / 2.72** / 3.29 | MOS($\uparrow$) | 3.56±0.08 |
| | UniAudio | | 2.63 / 2.44 / **3.66** | | **3.68**±**0.07** |
| Target Speaker Extraction | Wang et al. (2018) | PESQ($\uparrow$) / VISQOL($\uparrow$) / DNSMOS($\uparrow$) | **2.41 / 2.36** / 3.35 | MOS($\uparrow$) | 3.43±0.09 |
| | UniAudio | | 1.88 / 1.68 / **3.96** | | **3.72**±**0.06** |
| Singing Voice Synthesis | Liu et al. (2022) | - | - | MOS($\uparrow$) / SMOS($\uparrow$) | 3.94±0.02 / **4.05**±**0.06** |
| | UniAudio | | | | 4.08±0.04 / 4.04±0.05 |
| Text-to-Sound | Liu et al. (2023a) | FAD ($\downarrow$) / KL ($\downarrow$) | 4.93 / 2.6 | OVL ($\uparrow$) | 61.0±1.9 / 65.7±1.8 |
| | UniAudio | | **3.12** / 2.6 | / REL ($\uparrow$) | **61.9**±**1.9 / 66.1**±**1.5** |
| Text-to-Music | Copet et al. (2023) | FAD ($\downarrow$) / KL ($\downarrow$) | 4.52 / **1.4** | OVL ($\uparrow$) | **73.3**±**1.5 / 71.3**±**1.7** |
| | UniAudio | | **3.65** / 1.9 | / REL ($\uparrow$) | 67.9±1.7 / 70.0±1.5 |

are completed with 16 AMD MI200-64G GPUs. The detailed configuration of optimization is in Appendix A.3. To retain the performance of previous tasks during fine-tuning, following Conneau et al. (2020), the training data are re-sampled with respect to tasks with $\alpha = 0.05$. Top-k sampling is adopted consistently for inference, in which $k$ and the temperature are set to 30 and 0.8, respectively. As the global Transformer does not directly predict tokens, the sampling process only happens in the local Transformer inference.

**Evaluation:** For evaluation, most tasks are evaluated using both objective and subjective metrics [5]. Generally, for objective evaluation, Word Error Rate (WER) is used to evaluate the intelligibility of generated speech; Similarity Score (SIM) is for similarity in terms of speaker identity[6]; Perceptual Evaluation of Speech Quality (PESQ), VISQOL[7], DNSMOS [8] and Mel Cepstral Distortion (MCD) are signal-level quality metrics derived from human auditory research; Following (Copet et al., 2023), Fréchet Audio Distance (FAD), Kullback-Leiber (KL) Divergence, and Fréchet Distance (FD) are for audio fidelity and audio similarity; For subjective evaluation, MOS and SMOS are adopted to provide human-centric judgment for speech and sing related tasks. For text-to-sound and text-to-music tasks, we use overall quality (OVL), and relevance to the text input (REL) (Copet et al., 2023). Note all subjective results are obtained from Amazon Mechanical Turk[9] for fair comparison. Appendix F shows details of the subjective evaluation process.

## 3.2 THE RESULTS OF 7 GENERATIVE TASKS IN THE TRAINING STAGE

This section presents the overall evaluation results of the proposed UniAudio model over all 7 audio generation tasks during the training stage. A comprehensive comparison is conducted between UniAuduio and multiple prior works on each task, including not only the LM-based methods but also the diffusion model-based methods as well as other conventional audio generation methods. The detailed comparison is presented in Appendix B. We selected one of the most advanced prior work in each task and present the results in Table 2.

As suggested in Table 2, UniAudio is a versatile system that can handle all 7 audio generation tasks together and achieve competitive performance. Per subjective evaluation, UniAudio surpasses the baselines in 3 out of 6 tasks (TTS, VC, Sound); per objective evaluation, it achieves better results on 5 out of the 7 tasks except SVS and Music. We also find UniAudio under-perform on several metrics. UniAudio's subjective performance for SE and TSE is less competitive compared with its competitors, which is also observed in previous literature (Erdogan et al., 2023) that the signal-level evaluation metrics may not be suitable for LM-based generative methods. UniAudio cannot surpass the selected competitor (Copet et al., 2023) in the Text-to-Music task. We note that (Copet et al., 2023) is built with more private labeled data than our UniAudio.

---

[5]Following the setting of DiffSinger (Liu et al., 2022), SVS tasks don't report the objective results
[6]WER and SIM evaluation models follow Wang et al. (2023a)
[7]https://github.com/google/visqol
[8]https://github.com/microsoft/DNS-Challenge/tree/master/DNSMOS
[9]https://www.mturk.com/

## 3.3 THE RESULTS OF 4 GENERATIVE TASKS IN THE FINE-TUNING STAGE

Table 3: Performance evaluation for UniAudio and selected prior works in the fine-tuning stage

| Task | Model | Evaluation Metrics | Results |
|------|-------|--------------------|---------|
| Audio Edit | AUDIT (Wang et al., 2023d)
UniAudio | FD ($\downarrow$) / KL ($\downarrow$) | 20.78 / 0.86
**17.78 / 0.77** |
| Speech Dereverb. | SGMSE+ Richter et al. (2023)
UniAudio | PESQ($\uparrow$) / DNSMOS($\uparrow$) | **2.87** / 3.42
2.13 / **3.51** |
| Instructed TTS | GroundTruth
UniAudio | MOS($\uparrow$) / SMOS($\uparrow$) | **3.77±0.07 / 3.85±0.08**
3.61±0.09 / 3.71±0.09 |
| Speech Edit | TTS system regeneration
UniAudio | MCD($\downarrow$) / MOS($\uparrow$) | 6.98 / 3.69±0.08
**5.12 / 3.82±0.06** |

As UniAudio is designed to continuously support new audio generation tasks, this section reports UniAudio's performance on unseen tasks. The model is obtained by fine-tuning over 4 new tasks jointly and the results are presented in Table 3. Similar to section 3.2, for each task, we compare UniAudio's performance with one selected prior work and report the detailed results in Appendix B.

As shown in Table 3, the fine-tuned UniAudio model surpasses its baselines in audio edit and speech dereverberation and is approaching the ground-truth quality in the Instructed TTS task. For speech editing, UniAudio shows considerable improvement compared to generating the whole sentence.

## 3.4 ABLATION STUDY

### 3.4.1 BENEFIT OF BUILDING UNIFIED AUDIO GENERATION MODEL

To further validate our claim that building a unified model for all 11 audio generation tasks is promising and beneficial, more ablation studies are conducted. In Appendix C.1, we demonstrate that the joint-trained UniAudio model consistently outperforms the models that are trained for each specific task[10], regardless they are included in the training stage or the fine-tuning stage. In Appendix C.2, we additionally validate that fine-tuning over the 4 new audio generation tasks does not affect UniAudio's performance on the original 7 tasks. In Appendix C.3, we observe that UniAudio can consistently benefit from increased training data volume of each task, which provides another reason to build universal audio generation models: these models are easier to scale up as the data collection is more feasible. We provide more discussion in Appendix D about the effectiveness of building a universal audio generation model.

### 3.4.2 THE EFFECTIVENESS OF MULTI-SCALE TRANSFORMER MODEL

As in section 2.3, the adoption of neural codecs has become a popular choice of LLM-based audio generation but causes an overly long sequence issue that needs further consideration. This section compares the proposed multi-scale Transformer with four representative approaches in this field: Flattening Prediction (*e.g.* SPEARTTS (Kharitonov et al., 2023)), Coarse first prediction (*e.g.* VALL-E (Wang et al., 2023a)), Parallel prediction (*e.g.* AudioGen (Kreuk et al., 2022)), and Delay prediction (*e.g.* MusicGen (Copet et al., 2023)). Figure 2 illustrates the prediction order of these five architectures. Experiments are conducted on text-to-speech and text-to-music tasks and the results are reported in Table 4 and 5 respectively [11].

**Auto-Regression and Performance:** Among all 4 baselines aforementioned, Copet et al. (2023) claims that the flattening method provides the best audio generation quality. they further claim that the superior performance of flattening prediction is mainly attributed to the auto-regressive property; the other three methods do not reserve this property as the concurrent prediction is introduced (see Fig. 2). Under the scenario of codec adoption, we reinterpret the auto-regressive property as: current token prediction is based on all tokens of previous frames and the previous tokens within the current frame, or formally, the prediction of the current token $z_t^k$ is based on tokens: $\{z_{t'}^{k'}|t' < t\} \cup \{z_{t'}^{k'}|t' = t, k' < k\}$. With this definition, we claim that the proposed multi-scale transformer is also auto-regressive.

Aligned with Copet et al. (2023), our experiments also validate the importance of the auto-regressive property. As in Table 4 and 5, flattening prediction brings better generation quality than parallel,

---

[10]Note the task-specific models are built with the corresponding subset of the training data.

[11]Results are based on unofficial implementations.

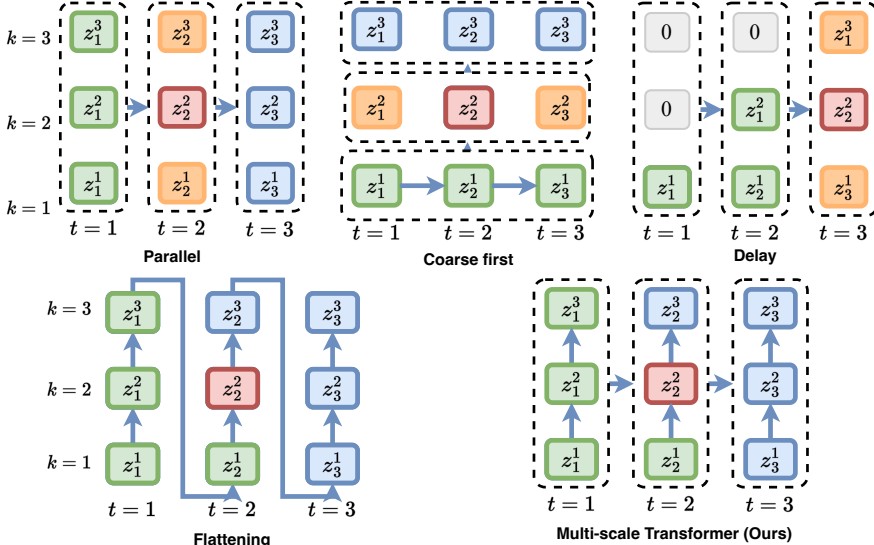

Figure 2: Order of token prediction for 4 representative methods in audio generation (Copet et al., 2023) and the proposed multi-scale Transformer. Assume $n_q = 3$ and $T = 3$. Current token prediction (red) is conditioned on prior tokens (in green). Tokens in orange are concurrently predicted with the current token. 0 is a special token indicating empty positions in the delay prediction.

Table 4: Model comparison among Coarse first, Flattening, Parallel, delay prediction, and multi-scale Transformer. Experiments were conducted on the LibriTTS. GPU memory and training time are obtained by a 20-second audio (average of 100 trials). All models have a similar parameter budget.

| Structure | $n_q$ | MOS ($\uparrow$) | MCD ($\downarrow$) | GPU Mem. (GB) | Time (s) / Iter. |
|---|---|---|---|---|---|
| Coarse first | 8 | 3.48±0.05 | 7.37 | 18.7 | 0.58 |
| Parallel | 3 | 3.14±0.07 | 7.89 | 13.56 | 0.53 |
| Delay | 3 | 3.48±0.05 | 6.95 | 13.65 | 0.59 |
| Flattening | 3 | 3.80±0.09 | 6.56 | 36.7 | 1.63 |
| Multi-Scale Transformer (ours) | 3 | 3.77±0.05 | 6.52 | 19.4 | 0.73 |
| Multi-Scale Transformer (ours) | 8 | 3.84±0.06 | 6.27 | 24.0 | 1.10 |

coarse first, and delay prediction. Additionally, with the same auto-regressive property, our proposed multi-scale transformer achieves a comparable performance with flattening prediction in terms of generation quality, which, again, validates the importance of auto-regression.

**Efficiency:** Besides generation quality, efficiency is a major concern of audio generation. Although with the auto-regressive property, the flattening prediction is sub-optimal in terms of efficiency: the modeling is based on the $T \times n_q$ long sequence, which has a space complexity of $O((T * n_q)^2)$ in self-attention. As increasing $n_q$ gives higher reconstruction quality at the cost of longer sequences and more computation, this issue becomes more severe when a larger $n_q$ is adopted. Since the sequence length grows proportionally with $n_q$, we experimentally find it difficult to train with $n_q \geq 4$. By contrast, the proposed multi-scale Transformer distributes the inter- and intra-frame modeling to the global and local sub-modules respectively, which thus alleviates the space complexity to $O(T *^2)$. Finally, without the requirement of auto-regression, methods like parallel, coarse first, and delay predictions achieve better efficiency due to the adoption of concurrent predictions. Since the space complexity is independent to $n_q$, training a larger $n_q$ with the multi-scale transformer is then feasible.

Experimentally, the proposed multi-scale transformer considerably reduces the time and memory cost compared with the flatting prediction. It still costs more time and memory compared with the other three baselines.

Based on the observations above, we claim that the proposed multi-scale transformer is an auto-regressive architecture that achieves a better trade-off between generation quality and efficiency.

## 4 RELATED WORKS

This work is an attempt to achieve universal audio generation through LLM-based techniques. There is a long research history for many audio generation tasks. Conventionally, the design of these tasks

Table 5: The ablation study to explore the effectiveness of our proposed multi-scale transformer. Experiments were conducted on text-to-music tasks with the Million Song dataset.

| Structure | $n_q$ | FAD ($\downarrow$) | KL ($\downarrow$) | OVL. ($\uparrow$) | REL. ($\uparrow$) |
|---|---|---|---|---|---|
| Parallel | 3 | 6.92 | 2.36 | 60.4±2.3 | 61.3±1.5 |
| Delay | 3 | 6.07 | 2.23 | 62.8±1.9 | 63.9±1.6 |
| Flatten | 3 | 5.18 | 1.83 | 64.8±1.8 | 65.2±2.0 |
| Multi-Scale Transformer (ours) | 3 | 5.24 | 1.80 | 64.4±2.1 | 66.2±2.4 |

heavily leverages the domain knowledge of each specific task, and their workflows are distinctive from each other: For tasks like TTS, SE, TSE, TT-Music, VC, S-Edit, SD, SVS, (1) their neural network architectures are based on Transformer (Ren et al., 2020) or others (Oord et al., 2016; Luo & Mesgarani, 2019); (2) their training objectives can be either in time-domain (Luo & Mesgarani, 2019), frequency-domain (Yu et al., 2017) or others (Gu et al., 2021; Shen et al., 2023); (3) their designs are inspired by and derived from linguistics and phonetics (Zen et al., 2013), signal processing (Griffin & Lim, 1984), auditory perception (Shadle & Damper, 2001) and machine learning (Wang et al., 2016) research, etc; (4) they use different generative models, such as diffusion model (Shen et al., 2023; Wang et al., 2023b), flow (Le et al., 2023), Seq2Seq (Ren et al., 2020; Liu et al., 2021).

The prosperity of LLM techniques (Radford et al., 2019; OpenAI, 2023) significantly promotes progress in audio generation research in several directions. First, the large language models, along with the prompt methods, inspired multiple emergent audio generation tasks that are based on textual instruction or descriptions from humans, such as Instruct-TTS (Yang et al., 2023a), Text-to-sound (Kreuk et al., 2022; Huang et al., 2023a) and text-to-music Copet et al. (2023); Agostinelli et al. (2023). Second, besides the text, audio can also be tokenized as discrete sequences (Zeghidour et al., 2021; Défossez et al., 2022; Kumar et al., 2023) that can be further processed by LMs. LM-based audio generative models then show superior capability in generalization towards unseen speakers (Wang et al., 2023a), low resources (Kharitonov et al., 2023) and multilingual (Zhang et al., 2023) scenarios. These methods also achieve state-of-the-art results in overall performance within their own scopes. Finally, the LM-like model can be further combined with existing generative models (e.g., diffusion models Rombach et al. (2022)) to obtain improved generation quality.

It is laborious to handle each audio generation task case-by-case, especially when considering the data shortage as well as the emergent and varying needs in this area. Alternatively, building a universal audio generation model is a promising and practical paradigm. Given the rapid progress in audio generation research, recent designs of audio generation, including LM-based ones, tend to support multiple audio generation tasks simultaneously. Some pioneer works (Wang et al., 2023c; Le et al., 2023; Shen et al., 2023; Liu et al., 2023b; Jiang et al., 2023) clearly consider supporting multiple tasks as a key strength; the designs of other prior works (Borsos et al., 2023; Kharitonov et al., 2023; Shen et al., 2023) do show the potential to generate audio in a broader sense than what they originally claim. Following these pioneering research works, UniAudio supports an extended coverage of 11 audio generation tasks in a unified LM-based model.

## 5  LIMITATION

Not all known audio generation tasks are included in the proposed UniAudio, such as noise removal, noise speech edit (Wang et al., 2023c) and speech-to-speech translation (Rubenstein et al., 2023; Barrault et al., 2023). All new tasks added in fine-tuning are formulated with the known modalities in the training stage; Introducing new modalities during fine-tuning is unexplored in this work. Current UniAudio considers neither unlabeled data nor domain-specific foundation models, which can possibly further improve the overall performance. The samples generated by UniAudio are not guaranteed in quality and may contain errors.

## 6  CONCLUSION

To handle the emergent and varying needs in audio generation, this work is an attempt to achieve universal audio generation. UniAudio is proposed as a unified LM-based generative model that supports 11 different audio generation tasks. In experiments, the proposed UniAudio provides competitive performance on all 11 tasks. It also empirically demonstrates the capability of continuously integrating unseen audio generation tasks. Demo and code are released, in the hope that UniAudio can become a foundation model for universal audio generation in further research.

## 7 ETHICAL STATEMENT

We are delving into the revolutionary field of generating diverse audio using large language model techniques. We find ourselves at the confluence of innovation and responsibility. It is imperative to acknowledge the ethical dimensions of our work and ensure that our contributions are employed for the betterment of society.

**Being Open:** As we advance in this domain, it's crucial to ensure that the benefits of this technology are widespread and not limited to a privileged few. Our code is released publicly along with this submission to ensure equal access for each person. All experiments are based on open-accessible datasets that allow research-oriented comparison and reproduction.

**Avoid Misuse:** While our model can produce a myriad of audio content ranging from music to speech, there's potential for misuse in the generation of misinformation, deepfake audio, or any harmful content. We advocate for adopting our code and model responsibly, with full respect to individual privacy and observance of regulations. Concerning the potential misuse of our model, checkpoints will not be released.

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
