# OpenReview forum: "UniAudio: An  Audio Foundation Model Toward Universal Audio Generation"
_ICLR.cc/2024/Conference — Submitted to ICLR 2024_

### Official Review · Reviewer_gngw · 2023-10-30

**Soundness:** 4 excellent
**Presentation:** 4 excellent
**Contribution:** 4 excellent
**Rating:** 10
**Confidence:** 3

**Summary:**

In this paper, the authors train a single multi-scale transformer to perform multiple different audio tasks. They demonstrate that, by learning these tasks together, they get a performance boost for this model, and achieve state of the art results on multiple tasks (against specialist models). They show that the existing model can be fine tuned to perform well on novel tasks. In addition, they describe how to translate multiple input modalities into a universal data space with a neural codec to allow this to happen, and discuss the merits and detriments of this system against other approaches.

**Strengths:**

* All data used is public, so results can be reproduced
* Compute required is not out of the realms of most academic institutions (16 x MI200-64G GPUs ~ $64k) - particularly if, as is suggested, this model is taken and simply fine tuned upon
* I am not aware of another model which has covered all of these tasks nor comprehensively demonstrated that combining different input modalities benefits each task; the authors show the benefit of training on multiple modalities by training their model on just the one task, and then again on all task, and by comparison with the current sota (generally single-task) model
* Care and attention has been taken to compare with recent sota models in each domain, and to perform both objective and subjective evaluations. In addition, an anonymised website of audio generations was provided

**Weaknesses:**

I would like to emphasise that all the following are suggestions for discussion / future work - IMHO this contribution is more than sufficient for publication in its current form.
* Reliant on the input representation - any losses incurred by input representation cannot be modelled, and if a new representation were to be used, a new model must be trained
* As noted in the limitation section, there's no demonstration of fine tuning to an unknown modality (which would require a new special modality token to be included in the vocabulary)
* The evaluation is so wide ranging it's difficult to parse in tables. Perhaps a bar chart or plot could improve the interpretation of relative performance with sota benchmarks?
* Only amazon turk was used for evaluation - a broader human evaluation with subject experts would be very interesting

**Questions:**

* It is not explained exactly why the 4 tasks were selected for the fine tuning study, was there any reason?
* Is there any reason that the process could not be end to end (i.e. must the neural codec be learned prior and fixed)?

---

> ### Author Response · Authors · 2023-11-23
> **Response to Reviewer gngw**
>
> We thank the reviewer for recognizing our contributions. We do appreciate the constructive comments the reviewer provided to us to
> further improve our paper. We are delighted to have the following discussion with the reviewer.
>
> **Q1:**  Reliant on the input representation - any losses incurred by input representation cannot be modelled, and if a new representation
> were to be used, a new model must be trained.
>
> **A:**  We are happy to have this discussion.
>
> (1) Theoretically, any information that is not retained by the input models (T5, mHuBERT, Codec) will not be modeled by UniAudio.
> However, we attempt to convince the reviewer that, the input models adopted in our experiments are exhaustively validated in downstream tasks (thanks to the community efforts) and should be able to reserve most of the necessary information.
>
> (2) UniAudio is exactly built upon these input models and needs to be trained again if the input models are changed. However, all input
> models adopted in this work are publicly available and our training code is also released to help our users. The reproducibility of our work is secured.
>
> **Q2:** As noted in the limitation section, there’s no demonstration of fine-tuning to an unknown modality (which would require a new
> special modality token to be included in the vocabulary).
>
> **A:** We thank the reviewer for reading our paper carefully. We agree with the reviewer that introducing new modalities is interesting and
> plan to address this topic in future work.
>
> (1) Currently, UniAudio supports multiple modalities (phone, text, codec, MIDI, semantic), which is probably sufficient for most users to
> define their new audio generation tasks.
>
> (2) By extending the vocabulary and adding the data from new modalities, UniAudio can possibly support more modalities.
>
> **Q3:** The evaluation is so wide ranging it’s difficult to parse in tables. Perhaps a bar chart or plot could improve the interpretation of
> relative performance with sota benchmarks?
>
> **Revision:** We appreciate this comment. Following this instruction, we revised our tables in the main content and added multiple bar
> charts in the appendix to present our experimental results.
>
> **Q4:** Only amazon turk was used for evaluation - a broader human evaluation with subject experts would be very interesting
>
> **A:** Thanks for this suggestion. We agree with the reviewer that human evaluation of AI-generated content is a popular open problem.
> We are willing to include more professional raters in our evaluation procedure and share more insights into how the generated content can better human benefits.
>
> **Q5:** It is not explained exactly why the 4 tasks were selected for the fine-tuning study, was there any reason?
>
> **A:** We thank the reviewer for this question on our experimental setup.
>
> (1) We clarify that we split 7 tasks for first-stage training and 4 tasks for fine-tuning to verify that the proposed UniAudio can act as a
> foundation model and continuously support new audio generation tasks. Instead, all 11 tasks can be possibly trained together from scratch.
>
> (2) When splitting the tasks, we intend to ensure all modalities have been learned during the first-stage training over the 7 tasks.
>
> (3) The 4 tasks for fine-tuning are comparatively small in data volume, which is well aligned with the situation that our users intend to
> build a new audio generation task but do not have large-scale data in the beginning.
>
> **Q6:** Is there any reason that the process could not be end to end (i.e. must the neural codec be learned prior and fixed)?
>
> **A:** We agree with the reviewer that building a universal audio generation model in a more end-to-end paradigm is promising and beneficial.  We are willing to work in this direction. However, the following problems could be challenging.
>
> (1) **Input:** Some necessary prior knowledge encoded in the input models can hardly be learned by training an audio generation model.
> E.g., the textual semantics provided by the T5 model.
>
> (2) **Output:** We consider our method as LM-based, which is featured by discrete unit prediction. Getting rid of audio tokenization (e.g.,
> audio codec) can possibly change the predicting target to continuous representations and is not aligned with the LM-like paradigm. Also,
> predicting the continuous representations can possibly encounter a larger dynamic range than predicting the discrete units, which may
> pose additional technical challenges.
> (3) **Speed:** If we directly generate waveform like the previous work by Wavenet (van den et al, 2016), the sequence would be very long, the prediction space would be very large, and the inference speed would be extremely slow.
>
> Reference: van den et al, WaveNet: A Generative Model for Raw Audio, in InterSpeech 2016

---

### Official Review · Reviewer_EAVx · 2023-11-01

**Soundness:** 3 good
**Presentation:** 3 good
**Contribution:** 3 good
**Rating:** 5
**Confidence:** 4

**Summary:**

- The authors present a foundation model for audio generation and editing which encompasses 11 tasks spanning multiple modalities of audio such as speech, music, and general sounds.
- The paper proposes a different approach to modeling audio tokenized into discrete codes using audio codecs. The proposed architecture is a single multi-scale transformer which involves a global transformer which operates on a summarized form of each audio frame, and a local transformer which performs autoregressive generation of codes within an audio frame. The design is proposed as a way to alleviate training transformers on very long sequences of flattened audio codes (SPEAR-TTS), or having train an autoregressive transformer on one level of codes and another non-autoregressive transformer on the remaining level of audio codes (VALL-E).
- The training process involves 7 different tasks with various task indicator tokens being used. The remaining tasks are incorporated in a fine-tuning phase post training of the base model.
- The authors perform evaluation using subjective and objective metrics and compare against various other models which may be specialized for each task. The results indicate that UniAudio is competitive against all the baselines.

**Strengths:**

- As the authors show in the first table, this system is the first to tackle so many tasks in audio generation using a single backbone with only additional fine-tuning.
- The proposed model architecture is different from those used in other audio generation models and shows some gains. The comparison is not necessarily fair because of differences in training data, but it does give the community a new option to consider while building audio generation models.
- Results for speech data and denoising are decent.

**Weaknesses:**

- The paper is a very dense read with some details relegated to the appendix while some details difficult to glean from the text. For example, the details about the audio tokenizer is moved to the appendix and no specifics are mentioned in the main text. Also, the text and the figure in the model architecture section are a little difficult to understand. I think I got it after reading through a few times, but adding some annotations to the figure and its caption will greatly improve readability.
- While the results for speech are decent, I found some issues w.r.t words being skipped. Also, the results on text to music/audio are pretty poor. The authors have not discussed any of the limitations in terms of quality of the generated audio despite the model being competitive with current state-of-the-art.
- The paper does not offer too much in terms of insights. Not to take away from the effort it takes to setup such a large scale experiment, but my main takeaways from the paper are as follows:
  1. Existing audio codecs are not suitable for wide range of tasks, so one needs to train them on more diverse data.
  2. As long as we collect 100k+ hours of audio data we can achieve such performance using the multi-scale transformer.

  The multi-scale transformer architecture is definitely interesting to the community but the architecture contribution is not well ablated, and in my opinion, that is the most important part of the paper.
- Minor correction: In Table 3, subjective and objective metrics column headers should be interchanged.

**Questions:**

- Wonder if the authors can add any ablations that answer questions related to the effect of data quantity?
- Did the authors consider using DAC as the audio codec? Those models are trained on a more diverse dataset and also have other improvements over EnCodec.

---

> ### Author Response · Authors · 2023-11-23
> **Response to Reviewer EAVx**
>
> We thank the reviewer for recognizing our contributions. We do appreciate the constructive comments the reviewer provided to us to
> further improve our paper. We are delighted to have the following discussion with the reviewer.
>
> **Q1:** The paper is a very dense read with some details relegated to the appendix while some details difficult to glean from the text. For
> example, the details about the audio tokenizer is moved to the appendix and no specifics are mentioned in the main text. Also, the text
> and the figure in the model architecture section are a little difficult to understand. I think I got it after reading through a few times, but
> adding some annotations to the figure and its caption will greatly improve readability.
>
> **Revision:**  We do appreciate the reviewer’s patience and apologize for the confusion caused by our interpretation. We do the following
> revisions to improve the readability:
>
> (1) We checked each appendix of the paper and made sure all appendices were properly mentioned in the main context, including our
> building process of the codec model.
>
> (2) We revised the diagram in the model architecture Section.
>
> (3) We changed some tables in the main content into diagrams for ease of understanding.
>
> (4) We double-checked all the text and annotations in the figures.
>
> **Q2 :** While the results for speech are decent, I found some issues w.r.t words being skipped. Also, the results on text to music/audio
> are pretty poor. The authors have not discussed any of the limitations in terms of quality of the generated audio despite the model being
> competitive with current state-of-the-art.
>
> **A:** We do appreciate these constructive comments. Please allow us to clarify as follows:
>
> (1) The samples in our demo pages are randomly selected and are reasonable to contain errors, including errors like skipping some words. We share that, we do not intend to exclude these errors to faithfully reflect our system’s performance. We acknowledge that UniAudio may generate some sub-optimal samples. We attempt to convince the reviewer that, the quantitative evaluation results of UniAudio could effectively reflect UniAudio’s performance.
>
> (2) We agree with the reviewer that our results on the text-to-music task do not outperform the strong prior work MusicGen. However, we encourage the reviewer to check that:
>
> (i) In Appendix B.4 which present our detailed results on text-to-music task, UniAudio outperforms many prior works except MusicGen
> and Noise2Music.
>
> (ii) Both MusicGen (20khrs) and Noise2Music (280khrs) adopted much more labeled data than our UniAudio (8khrs). Their data is private.
> The Million Song Dataset adopted in this work is the largest text-music dataset we can publicly access.
>
> (ii) Even being inferior to some prior works, we attempt to convince the reviewer that our results on the text-to-music task are self-consistent  to prove the benefit of building a unified audio generation model.
>
> (3) **Our Revision:**  We do appreciate the reviewer’s comments that the quality risk of our model should be clearly stated. In the
> limitation section, we state *The samples generated by UniAudio are not guaranteed in quality and may contain errors.*

---

> > ### Author Response · Authors · 2023-11-23
> > **Response to Reviewer EAVx part 2**
> >
> > **Q3:** The paper does not offer too much in terms of insights. Not to take away from the effort it takes to setup such a large scale experiment, but my main takeaways from the paper are as follows:
> >
> > **A:** We do agree with the insights summarized by the reviewer. Please allow us to additionally mention:
> >
> > (1) With the goal of achieving universal audio generation, UniAudio is a unified model that supports 11 audio generation tasks. Following the prior works that support multiple audio generation tasks, UniAudio achieves an impressive coverage extension.
> >
> > (2) This paper provides clear evidence to prove that building a universal model for audio generation is technically feasible and beneficial to nearly all 11 tasks included.
> >
> > (3) We figure out that human effort and resource input can be saved by building the universal models. Additionally, pioneer research in NLP also proves that universal generation is possibly a good way to achieve more advanced intelligence.
> >
> > (4) We mentioned that the proposed universal audio generation model can become another kind of foundation model in the audio community. It provides one possibility of what the large models in the audio modality would be like.
> >
> > **Q4:** The multi-scale transformer architecture is definitely interesting to the community but the architecture contribution is not well ablated, and in my opinion, that is the most important part of the paper.
> >
> > **A:** We appreciate that the reviewer is interested in our proposed architecture. We updated Section 3.4.2 with more supplementary experiments and discussions to compare the proposed multi-scale transformer with many advanced architectures presented in the literature. The experiments are conducted on both speech and non-speech tasks. The results still validate that the proposed architecture is strong in both performance and efficiency.
> >
> > **Q5:** Wonder if the authors can add any ablations that answer questions related to the effect of data quantity?
> >
> > **A:** We agree with the reviewer that this comparison is beneficial. These experimental results are newly added in Appendix C.3 Figure 4.
> >
> > **Q6:** Did the authors consider using DAC as the audio codec? Those models are trained on a more diverse dataset and also have other improvements over EnCodec.
> >
> > **A:** We are delighted to see that the recently published descript-audio-codec (DAC) achieved impressive improvements under many scenarios. We understand the importance of codec quality and are glad to adopt this codec model into UniAudio. We will cite this paper and integrate DAC into our open-source UniAudio code so that other readers can use DAC as an audio codec model to train their model.
> >
> > **Our Revision:** We cite the DAC paper.
> >
> > Reference:
> >
> > Copet et al, Simple and Controllable Music Generation, in NeurIPS 2023

---

### Official Review · Reviewer_kfiC · 2023-11-05

**Soundness:** 3 good
**Presentation:** 2 fair
**Contribution:** 1 poor
**Rating:** 1
**Confidence:** 5

**Summary:**

This paper presents a LM-like audio model, that is able to conduct multiple (11) audio generation tasks. Audio is tokenized with a EnCodec-like quantizer to fit for LM. Multi-scale Transformer is used for handling multiple tokens per audio frames. Experiment was conducted by training on 165K hours audio data.

**Strengths:**

The overall approach is sound. It's great to see a single model can handle so many different audio generation tasks.

**Weaknesses:**

1. The novelty of this paper is very limited. All the components used in the paper are previously existing. The main contribution of this paper is to train a multi-task model with a mixture of multiple datasets.
2. The overall contribution to the research community seems very limited. There is no much insights can be drawn from this paper to benefit the community. There is no ablation study conducted. It looks more of an engineering work than a research work.
3. The presentation of this paper needs improvement. It seems presented in an eye-catching, but misleading way. For example, it's improper to put Table 1 as the first thing in the content of this paper, because it lacks context and is confusing. Even worse, it's not a fair comparison -- the cited papers didn't include experiments on specific tasks doesn't mean that the methods presented in those papers are incapable of those tasks.

**Questions:**

None.

**Details Of Ethics Concerns:**

See the 3rd item in "weaknesses".

---

> ### Author Response · Authors · 2023-11-23
> **Response to Reviewer kfiC**
>
> We thank the reviewer for recognizing our contributions. We do appreciate the constructive comments the reviewer provided to us to
> further improve our paper. We are delighted to have the following discussion with the reviewer.
>
> **Q1:** The novelty of this paper is very limited. All the components used in the paper are previously existing. The main contribution of
> this paper is to train a multi-task model with a mixture of multiple datasets.
>
> **A:**  We thank this comment and are delighted to have the following discussion with the reviewer in terms of our contributions:
>
> (1) **Contribution on independent tasks:** Although rapid advances in LM-based audio generation, this technique is not widely adopted
> in some tasks. We attempt to convince the reviewer that applying the LM-based audio generation technique to tasks like instructed textto-speech and speed editing is novel.
>
> (2) **Contribution on unified tasks definition:** We attempt to reach a consensus with the reviewer that the proposed unified task
> formulation is of sufficient contribution. Considering the distinctions among all these audio generation tasks, such as the model backbones, training objective, generative paradigm, and task-specific domain knowledge, proposing a unified formulation for all 11 tasks is of sufficient contribution.
>
> (3) **Contribution on architecture:** We hope the reviewer can acknowledge our contribution to model architecture design. The proposed
> multi-scale transformer architecture is specifically designed for long audio token sequences with neural codec. The proposed architecture fully preserves the inter- and intra-frame auto-regressive property. The performance and efficiency of the proposed architecture are validated by the experiments and ablation study (see section 3.4.2).
>
> (4) **Contribution on experiments:** We also encourage the reviewer to check our refined experimental section: we experimentally prove
> that building a unified audio generation model is feasible and even beneficial to all of the tasks included. We further validate that the proposed UniAudio model can easily be generalized to unseen audio generation tasks with simple fine-tuning. These experimental results support our claim that UniAudio can be a foundation model that supports emergent audio generation needs in future research.
>
> **Q2:** The overall contribution to the research community seems very limited. There is no much insights can be drawn from this paper to benefit the community. There is no ablation study conducted. It looks more of an engineering work than a research work.
>
> **A:** We thank this comment and deeply apologize for the confusion caused by our interpretation. Please allow us to clarify as follows:
>
> (1) **Contributions and Insights:** First, we encourage the reviewer to check the contributions we summarized in Q1. Secondly, we
> would like to share and discuss the insights as follows: (i) Compared with building independent models for each task and each domain,
> building a universal audio generation model can effectively save human efforts and resource input. (ii) The universal audio generation
> models can potentially serve as foundation models that can generalized to unseen audio generation tasks. (iii) The goal of this research,
> achieving universal generation, is aligned with the observations of the pioneer works (Wang et al, 2022, Sanh et al, 2022) in domains other than audio. These pioneer works demonstrate that multi-task learning leads to a more advanced understanding of the target domain and improves the generalization towards unseen tasks.
>
> (2) **Abalation Study:** We apologize that we did not make our ablation studies clearly presented in the original manuscript. We would
> like to refer the reviewer to section 3.4 for our ablation studies and corresponding analysis. In these studies, we validate that (i) building a
> unified audio generation model is beneficial to all audio generation tasks included and (2) the proposed multi-scale transformer architecture fully preserves the autoregressive property while maintaining efficiency.
>
> (3) **Research or Engineer Work:** We attempt to reach a consensus with the reviewer that our work has contributions to both research
> and engineering. Our research contributions have been summarized above. We hope the reviewer will agree that these items are of sufficient research value. We also appreciate the reviewer for recognizing our efforts in engineering. Also, We have released our code to share our engineering efforts with the community.

---

> > ### Author Response · Authors · 2023-11-23
> > **Response to Reviewer kfiC part 2**
> >
> > **Q3:** The presentation of this paper needs improvement. It seems presented in an eye-catching, but misleading way. For example, it’s improper to put Table 1 as the first thing in the content of this paper, because it lacks context and is confusing. Even worse, it’s not a fair comparison – the cited papers didn’t include experiments on specific tasks doesn’t mean that the methods presented in those papers are incapable of those tasks.
> >
> > **A:** We deeply apologize for presenting our work in that improper way and do accept the reviewer’s comments that Table 1 in the original manuscript lacks in acknowledging the contributions of the prior works. We do admit that prior works have already supported multiple audio generation tasks and UniAudio achieves a broader coverage in terms of tasks. To address the reviewer’s concern, we revised our manuscript as follows:
> >
> > (1) We removed Table 1 in the revised manuscript to avoid potentially over-claiming our contributions.
> >
> > (2) We revised the related work section as *Given the rapid progress in audio generation research, recent designs of audio generation, including LM-based ones, tend to support multiple audio generation tasks simultaneously. Some pioneer works clearly consider supporting multiple tasks as a key strength; the designs of other prior works do show the potential to generate audio in a broader sense than what they originally claim. Following these pioneering research works, UniAudio supports an extended coverage of 11 audio generation tasks in a unified LM-based model.*
> >
> > (3) We revised the limitation section to clearly state *Not all known audio generation tasks are included in the proposed UniAudio, such as noise removal, noise speech edit (Wang et. al, 2023), and speech-to-speech translation.*
> >
> > Reference:
> >
> > Wang et al, Generalization via Declarative Instructions on 1600+ NLP Tasks in EMNLP 2022.
> >
> > Sanh et al, Multitask Prompted Training Enables Zero-Shot Task Generalization, in ICLR 2022.
> >
> > Copet et al, Simple and Controllable Music Generation, in NeurIPS 2023.
> >
> > Wang et. al, SpeechX: Neural Codec Language Model as a Versatile Speech Transformer, arxiv, 2023.

---

### Official Review · Reviewer_dsre · 2023-11-06

**Soundness:** 3 good
**Presentation:** 3 good
**Contribution:** 2 fair
**Rating:** 5
**Confidence:** 3

**Summary:**

The paper presents a foundation model for audio generation capable of handling a number of different tasks, all of which output audio (including speech, environmental sounds, and music) conditioned on input from multiple modalities including text, audio, and MIDI.

The paper also introduces a two-level Transformer architecture where the top-level attention operates across audio *frames* (where each frame consists of multiple discrete tokens) and the bottom-level attention operates on the tokens within a frame.

**Strengths:**

I am very much in favor of the goal of the paper: to introduce a multitask foundation model for audio generation.  I am also totally on board with the proposed architecture; decomposing the quadratic attention into frame-level and token-level is an excellent idea for handling the longer sequence lengths that are necessary for freeform audio generation.

**Weaknesses:**

Two main areas of weakness:

1) Most of the evaluation tasks do not seem ideal for demonstrating the effectiveness of an architecture designed to enable long sequence lengths; other than text-to-music, the tasks are all quite local, with no real dependencies longer than a few seconds.  Notably, text-to-music is the task on which the model in the paper performs worst compared to baseline.  And the experimental evaluation of the architecture vs others consumes only a small section of the paper and is fairly inconclusive.

Essentially, the architecture seems overkill for most of these tasks, and for the task for which one might expect it to help most (text-to-music), possibly the smaller training data size prevents the model from taking advantage of its capability.

2) The paper does not convincingly demonstrate that training a single model on speech-, music-, and sound-generating tasks exhibits synergy across domains.  While for most of the evaluation tasks the multitask model outperforms the single task model, a) the difference in performance is small and b) it's not clear that the model really benefits from training on all of speech, music, and sound compared to a separate speech model, music model, and sound model as this experiment was not performed.  It's also not clear how much of the benefit depends on specifics of the datasets and tasks used here vs a more general principle; e.g. the overall amount of speech data here is much larger than the other domains and so a music task might benefit from training jointly with speech more than would be the case if the amount of music data were larger.

Overall, I'm just not satisfied that the main hypothesis of the paper -- training a model on all audio generation domains at once is better than training separate models for each domain -- is sufficiently backed up with experimental evidence.  The fact that the joint model outperforms state of the art on about half the tasks (and *not* the music tasks) corroborates my dissatisfaction here.

One other possible reason for training on multiple domains could be: only a small quantity of training data exists for certain domains.  However, for the case of foundation models this is certainly not true; it is usually *labels* that are in short supply, and raw audio -- speech, music, and sound -- exists in enormous quantity.

**Questions:**

1) Basically, convince me that training on speech helps with music, even if one has access to enormous unlabeled music datasets.  I'm totally willing to believe that training on all the speech tasks at once is helpful.

2) I do believe that the architecture proposed is going to outperform the comparison architectures for audio generation.  But the experiment demonstrating this isn't especially compelling; did you perform other experiments on the different architectures?

---

> ### Author Response · Authors · 2023-11-23
> **Response to Review dsre**
>
> We thank the reviewer for recognizing our contributions. We do appreciate the constructive comments the reviewer provided to us to
> further improve our paper. We are delighted to have the following discussion with the reviewer.
> **Q1:** Discussion on the potential adoption of unlabeled data and domain-specific models.
>
> **A:** We believe the reviewer is most interested in this direction so we reply to it first. We do agree with the reviewer the paper should
> consider the unlabeled data and domain-specific models. We summarize and respond to the reviewer’s comments in the following aspects.
>
> (1) **Unlabeled Data:** We appreciate and agree with the reviewer that the adoption of unlabeled data (e.g., unlabeled music) can lead to
> performance improvement. We then did experiments in this direction and our primary experiments support the reviewer’s comment (see
> Table 1 below). We do appreciate this inspiration.
>
> (2) **Unified vs. Domain-Specific:** We also agree with the reviewer that we haven’t presented comparative results between the proposed unified model and the domain-specific models (e.g., music-specific models). We attempt to convince the reviewer that foundation models of these two kinds have different features, serve different application scenarios, and thus should not necessarily be compared. We agree that, compared with the unified model, the domain-specific models may have more expertise in that domain and lead to better performance on domain-specific tasks. However, building domain-specific models for each domain also requires more effort and resource input. By contrast, the unified model is expected to obtain prior knowledge in multiple domains and can be more flexible in handling newly defined audio generation tasks. Thus, we believe both the universal and domain-specific foundation models are worth exploring.
>
> (3) **Our Revisions:** We thank the reviewer for this inspiration. To avoid potential over-claim or ambiguity, in the limitation section,
> we clearly state that *Current UniAudio considers neither unlabeled data nor domain-specific foundation models, which can possibly further improve the overall performance*; and in the experimental setup section, we state *UniAudio is built on labeled datasets*.
>
> Table 1: The ablation study to explore the influence of adding unlabeled music data on text-to-music tasks. With
> joint training paradigm: (1) text-to-music is trained with the 8khrs labeled data (Million Song dataset); (2) Unlabeled
> music prediction is trained as a mask-and-predict task and is based on the 8k hours of unlabeled music data (Luca
> et.al). The adoption of unlabeled music data does achieve performance improvement on text-to-music task.
>
> | Setting                                    | FAD ($\downarrow$) | KL ($\downarrow$) | OVL. ($\uparrow$) | REL. ($\uparrow$) |
> |--------------------------------------------|--------------------|-------------------|-------------------|-------------------|
> | Text-to-Music only                         | 5.24               | 1.80              | 64.4$\pm$2.1      | 66.2$\pm$2.4      |
> | Text-to-Music + Unlabeled music prediction | 3.95               | 1.85              | 65.4$\pm$1.6      | 68.8$\pm$1.5      |
>
>
> **Q2:** Most of the evaluation tasks do not seem ideal for demonstrating the effectiveness of an architecture designed to enable long sequence lengths.
>
> **A:** We thank the reviewer for reminding us to clarify UniAudio’s capability to generate long audio. We are delighted to discuss this with
> the reviewer as below.
>
> (1) We are happy to share that the proposed UniAudio can actually support to generate long audio. In our experimental setup, the
> global transformer supports 3,000 audio frames at maximum; the training target is truncated to 20 seconds, with the possibility of further
> increase.
>
> (2) We also agree with the reviewer that long-form audio generation is a promising direction to explore. However, supporting long-form
> generation is less claimed as a key strength of UniAudio. We would like to clarify that, although we propose the multi-scale transformer
> architecture to handle the overly-long sequences in audio generation, this issue is more raised from the adoption of the neural codec model and residual vector quantization (Zeghidour et al., 2021) than the original length of the audio.
>
> (3) **Our Revision:** We encourage the reviewer to check our demo page with more long-form examples. The newly updated examples
> have lengths of around 20 seconds, which is aligned with the common needs of audio generation.

---

> > ### Author Response · Authors · 2023-11-23
> > **Response to Review dsre part 2**
> >
> > **Q2:** Other than text-to-music, the tasks are all quite local, with no real dependencies longer than a few seconds.
> >
> > **A:**  We are happy to discuss the local dependency issue with the reviewer, and we do agree with the reviewer the proposed model should be able to handle both local and global dependency.
> >
> > (1) Among the 11 audio generation tasks supported by UniAudio, many tasks are naturally local and attempt to translate the input into output in a verbatim-like manner, such as speech enhancement, voice conversion, etc. As mentioned by the reviewer, the modeling of these tasks is mainly local.
> >
> > (2) We would also like to encourage the reviewer to check that, besides text-to-music, global modeling also broadly exists in other tasks, such as the enrollment speech in target speech extraction; speaker prompt in text-to-speech and voice conversion; the textual instructions in instruct TTS and textual descriptions in text-to-sound. We attempt to convince the reviewer that the proposed UniAudio can still handle the global dependency properly.
> >
> > **Q3:** Notably, text-to-music is the task on which the model in the paper performs worst compared to the baseline. The fact that the joint model outperforms the state-of-the-art on about half the tasks (and not the music tasks) corroborates my dissatisfaction here.
> >
> > **A:** We thank the reviewer for checking our results carefully. We admit our results on the text-to-music task do not outperform the strong prior work MusicGen. However, we encourage the reviewer to further check that:
> >
> > (1) In Appendix B.4 which present our detailed results on text-to-music task, UniAudio outperforms many prior works except MusicGen  and Noise2Music.
> >
> > (2) Both MusicGen (20khrs) and Noise2Music (280khrs) adopted much more labeled data than our UniAudio (8khrs). Their data is private. The Million Song Dataset adopted in this work is the largest text-music dataset we can publicly access.
> >
> > (3) Even being inferior to some prior works, we attempt to convince that our results on text-to-music task are self-consistent. We refer the reviewer to Q5 for further discussion.
> >
> > **Q4:** The experimental evaluation of the architecture vs others consumes only a small section of the paper and is fairly inconclusive.
> >
> > **A:** We thank the reviewer for raising this issue and do agree the results presented in our initial manuscript are not conclusive. Following this instruction, we conducted a more comprehensive comparison between the proposed multi-scale Transformer and 4 other architectures listed in Copet et al, (2023). We refer the reviewer to check our revised section 3.4.2 for details. We attempt to convince the reviewer that the proposed architecture fully preserves the autoregressive property while maintaining efficiency.

---

> > > ### Author Response · Authors · 2023-11-23
> > > **Response to Review dsre part 3**
> > >
> > > **Q5:** Discussion on the cross-task and cross-domain benefits of building a unified audio generation model.
> > >
> > > **A:** We are delighted that the reviewer reminds us to consider the benefits from both cross-task and cross-domain perspectives. We are happy to discuss with the reviewer that:
> > >
> > > **Cross-Task benefits:**  (Without the considerations of unlabeled data and domain-specific models) the original manuscript attempts to demonstrate that the proposed UniAudio achieves consistently better performance than its task-specific counterparts. Some of these improvements are even non-trivial, such as target Speaker Extraction, Text-to-Sound, Text-to-Music, Speech Dereverberation, and Audio Edit. Specifically, the results on text-to-music considerably improve due to the joint training (OVL 64.4− >67.9; REL 66.2− >70.0). We encourage the reviewer to check Appendix C.1 for details.
> > >
> > > **Cross-Domain benefits:** Thanks to the inspiration from the reviewer, we conducted primary experiments to show that the performance improvement can also be attributed to joint-domain training. Specifically, we design 4 settings and evaluate the results on the text-to-music tasks: (1) training UniAudio on a single text-to-music task; (2) training UniAudio on the text-to-music task and text-to-sound task; (3) training UniAudio on text-to-music and text-to-speech tasks; (4) training UniAudio on text-to-music, text-to-sound, and all of speech-related tasks. The experimental results are shown in Table 1. We can observe that adopting non-music data is consistently beneficial to all metrics except the KL metric.
> > >
> > > Table 2: The ablation study explores whether using more speech and sound data can bring improvement for text-tomusic tasks.
> > > | Data                   | FAD ($\downarrow$) | KL ($\downarrow$) | OVL. ($\uparrow$)     | REL. ($\uparrow$)     |
> > > |------------------------|--------------------|-------------------|-----------------------|-----------------------|
> > > | Music only             | 5.24               | 1.80    | 64.4$\pm$2.1          | 66.2$\pm$2.4          |
> > > | Music + Sound          | 4.35               | 1.93              | 65.8$\pm$1.9          | 66.5$\pm$2.3          |
> > > | Music + Speech         | 4.66               | 1.97              | 64.9$\pm$1.7          | 67.6$\pm$2.0          |
> > > | Music + Sound + Speech | 3.65      | 1.90              | 67.9$\pm$1.8 | 70.0$\pm$1.5 |
> > >
> > > Reference:
> > >
> > > Copet et al, Simple and Controllable Music Generation, in NeurIPS 2023
> > >
> > > Wang et al, Generalization via Declarative Instructions on 1600+ NLP Tasks in EMNLP 2022
> > >
> > > Sanh et al, Multitask Prompted Training Enables Zero-Shot Task Generalization, in ICLR 2022

---

### Author Response · Authors · 2023-11-23
**General Response**

We thank the meta-reviewer for organizing this helpful peer review stage. We thank all reviewers for their time, patience, and constructive
comments to help us improve our paper. We have revised our manuscript to respond to all reviewers’ comments in a one-by-one style.
Noticeable revisions are highlighted in blue text in the paper.
**We also encourage the reviewers to check the PDF version of our response letter in the supplementary material for better readability**.

Based on the helpful comments of the reviewers, the main revisions in this update are:

(1) We reorganized the experiment section by adding the ablation study part to better present our experimental analysis; we added an
extra appendix to systematically present our ablation study.

(2) We revised our discussion on model architecture to provide a more comprehensive comparison with prior works in terms of the model
architecture.

(3) We revised our related work section to better acknowledge the contributions of prior works.

(4) We revised our limitation section based on issues raised by our reviewers.

(5) We revised our tables and diagrams for better readability.

(6) We update the demo page to show UniAudio’s capability in long audio generation.

(7) We remove the content that can potentially over-claim our contributions.

Based on the revised manuscript, we attempt to reach a consensus with the reviewers about our main contributions:

(1) To achieve universal audio generation, UniAudio is a unified model that supports 11 audio generation tasks. Following the prior works
that support multiple audio generation tasks, UniAudio achieves an impressive coverage extension.

(2) The proposed UniAudio can support 11 audio generation tasks due to multiple novel considerations, such as (i) sequential representations. of audio and other input modalities; (ii) uniform formulation for LLM-based audio generation tasks; and (iii) efficient model architecture specifically designed for audio generation.

(3) In experiments, the overall performance of UniAudio is well validated. Further ablation studies demonstrate that (i) building a
joint-trained audio generation model is beneficial to nearly all audio generation tasks included; (ii) the proposed multi-scale transformer
architecture is effective in modeling the audio token sequences.

(4) Demo and code are released, in the hope that UniAudio can become a foundation model that supports emergent audio generation in
future research.

We would also like to discuss with the reviewers why achieving universal audio generation is of sufficient interest and why UniAudio can
potentially benefit the research community:

(1) Compared with building independent models for each task and each domain, building a universal audio generation model can effectively save human efforts and resource input.

(2) The universal audio generation models can potentially serve as foundation models that can generalized to unseen audio generation
tasks.

(3) The goal of this research, achieving universal generation, is aligned with the observations of the pioneer works (Wang et al, 2022, Sanh et al, 2022) in domains other than audio. These pioneer works demonstrate that multi-task learning leads to a more advanced understanding of the target domain and improves the generalization towards unseen tasks.

Reference:

Copet et al, Simple and Controllable Music Generation, in NeurIPS 2023.

Wang et al, Generalization via Declarative Instructions on 1600+ NLP Tasks in EMNLP 2022.

Sanh et al, Multitask Prompted Training Enables Zero-Shot Task Generalization, in ICLR 2022.

---

### Meta-Review · Area_Chair_37Zp · 2023-12-06

**Metareview:**

The authors propose an LLM-like approach to generate multiple types of audio (speech, sounds, music, singing) all using a single model. The idea is to tokenize audio and other modalities (phones, textual description, and audio context), concatenate the source and target as a sequence and perform next-token prediction using LLM. The authors also propose using a multi-scale Transformer to handle longer sequences (since the tokenizer is based on residual vector quantization), consisting of a global transformer for modeling “semantics” (inter-frame) and local Transformer for finer scale “acoustics” (intra-frame). Results show that across 11 tasks, the proposed model is moderately competitive, outperforming other approaches in some metrics for some tasks to prior works.

The reviewers agreed that the idea of using a foundational model that covers all audio related tasks is interesting. Furthermore, this is likely the first work that covers the proposed set of tasks covering the domains of speech and music. They also found the multi-scale transformer architecture to be quite interesting in the context of modeling longer sequences.

During the review, a few important concerns were also raised. While training on multiple tasks using a larger training set is in itself impressive, the results were not convincing enough to showcase the benefits of training on multiple tasks. The model works competitively on some tasks (speech related, e.g.), but not as well on some other (text-to-music, e.g.), which is inline with what one would expect when training on data from multiple tasks jointly. The multiscale transformer architecture is interesting, but this is based on prior work (Yu et al., 2023), and there weren’t enough ablations motivating the specific architectural choices. The reviewers were also unconvinced that the comparisons for specific tasks were always SOTA for the single-task setting.

The authors did a great job addressing a number of questions raised during the review / rebuttal phase, including additional experiments and results. But the fundamental concerns of clearly showing the benefits of training on the chosen set of tasks remain, especially on the harder tasks that need long context for generation, like music; and for the simpler tasks, the proposed architecture might be unnecessarily complex. The benefit, in the end, of having a moderately good “foundational” audio model, is not compelling enough.

**Justification For Why Not Higher Score:**

While the idea of building a foundational mode for audio is interesting, the comparisons, evaluation choices and the results on some of the chosen tasks are not convincing enough.

**Justification For Why Not Lower Score:**

N/A

---

### Decision · Program_Chairs · 2024-01-16

Reject